# Two Male-Specific Antimicrobial Peptides SCY2 and Scyreprocin as Crucial Molecules Participated in the Sperm Acrosome Reaction of Mud Crab *Scylla paramamosain*

**DOI:** 10.3390/ijms23063373

**Published:** 2022-03-21

**Authors:** Ying Yang, Fangyi Chen, Kun Qiao, Hua Zhang, Hui-Yun Chen, Ke-Jian Wang

**Affiliations:** 1State Key Laboratory of Marine Environmental Science, College of Ocean and Earth Sciences, Xiamen University, Xiamen 361102, China; yvonneyang0803@gmail.com (Y.Y.); chenfangyi@xmu.edu.cn (F.C.); qiaokun@xmu.edu.cn (K.Q.); zhanghua0209@stu.xmu.edu.cn (H.Z.); hychen@xmu.edu.cn (H.-Y.C.); 2State-Province Joint Engineering Laboratory of Marine Bioproducts and Technology, College of Ocean and Earth Sciences, Xiamen University, Xiamen 361102, China

**Keywords:** invertebrate, antimicrobial peptide (AMP), fertilization, sperm, acrosome reaction, progesterone, SCY2, scyreprocin

## Abstract

Antimicrobial peptides (AMPs) identified in the reproductive system of animals have been widely studied for their antimicrobial activity, but only a few studies have focused on their physiological roles. Our previous studies have revealed the in vitro antimicrobial activity of two male gonadal AMPs, SCY2 and scyreprocin, from mud crab Scylla paramamosain. Their physiological functions, however, remain a mystery. In this study, the two AMPs were found co-localized on the sperm apical cap. Meanwhile, progesterone was confirmed to induce acrosome reaction (AR) of mud crab sperm in vitro, which intrigued us to explore the roles of the AMPs and progesterone in AR. Results showed that the specific antibody blockade of scyreprocin inhibited the progesterone-induced AR without affecting intracellular Ca^2+^ homeostasis, while the blockade of SCY2 hindered the influx of Ca^2+^. We further showed that SCY2 could directly bind to Ca^2+^. Moreover, progesterone failed to induce AR when either scyreprocin or SCY2 function was deprived. Taken together, scyreprocin and SCY2 played a dual role in reproductive immunity and sperm AR. To our knowledge, this is the first report on the direct involvement of AMPs in sperm AR, which would expand the current understanding of the roles of AMPs in reproduction.

## 1. Introduction

Reproduction is a precisely regulated serial process in all animals, and the step that guarantees sperm–egg fusion is called acrosome reaction (AR) [1]. The sperm AR was first described in sea urchin by Dan [2]; afterwards, this profound structural change of sperm has been identified in many species. The acrosome, a specialized organelle located at the tip of the head of sperm, has been described in a diverse array of species, including Arthropoda (crabs, shrimps, etc.), Mollusca, Annelida, Echinoderma, Cephalochordata, Chordata, and some Vertebrata. In response to certain stimuli (hormones, alkaline environment, or physical contact with the egg envelope), the acrosome undergoes AR [2]. During AR, sperm experiences the protruding of the acrosome, exocytosis of the acrosome vesicle (AV) which releases factors that facilitate penetration of the vitelline coat, thus completing sperm–egg fusion. However, some taxa such as teleost fish have no sperm acrosome [3], whereas some species such as insects possess sperm acrosome but do not undergo AR to penetrate the egg coat [4].

The AR mechanism varies among different species due to their diverse gamete structures and sites of fertilization [5]. Sea urchin, a marine invertebrate executing external fertilization, has made a great contribution to our understanding of AR [6,7,8]. It is now clear that the AR of sea urchin is triggered by the binding of fucose homopolymers in egg jelly to its receptor (REJ-1) on sperm [9]. The AR-inducing substances (ARIS) differ among animal species, and progesterone (PG) is one of the common ARIS in different species. In situ study of mammalian sperm, AR is generally not feasible, and mouse sperm are often used as the model of choice. Cumulative data showed that the AR in mammalian sperm could be triggered by the alkaline environment in the female reproductive tract, zona pellucida, and directly activated by low molecular weight compounds such as PG [10,11,12]. In vitro investigations on the AR of human sperm also indicate that PG and neurotransmitters are necessary for AR [13,14]. It is widely accepted that AR requires Ca^2+^ influx [15] and the intracellular Ca^2+^ concentration ({Ca^2+^}_i_) is the most important factor regulating sperm activity and changes throughout all steps of sperm activity [16,17]. The sperm-specific ion channel, CatSper, was reported to be involved in PG sensation and Ca^2+^ influx during mammalian sperm AR [18,19]. The CatSper also contributes to the chemotaxis of sea urchin sperm [20] and is therefore considered to be a universal channel associated with sperm function. However, although a number of AR-associated molecules have been found in different species, most of them have not been clearly elucidated yet [21,22,23].

Mud crab, *Scylla paramamosain*, is a typical marine arthropod and an important economic aquatic species with high commercial value in southeast China and Asian countries. Mud crabs molt over 20 times in their lifetime. The last molt (reproductive molt) of the female crabs initiates the mating process and stimulates ovarian development including oogenesis. After mating, sperm are transferred into the female spermathecae where they stay for one or several months until eggs maturity and then released at ovulation to complete the sperm–egg fusion [24]. Mud crabs undergo internal fertilization and produce alflagellate sperm [25]. To our knowledge, some key AR-associated molecules such as CatSper have not been reported in mud crab and we also failed to identify it even after a sincere analysis of the mud crab genome. Thus, mud crab may have a different AR molecular basis due to the lack of certain key AR-associated components that have been identified in other species. 

To date, a number of antimicrobial peptides (AMPs) have been identified in the male reproductive system in different species. These reproduction-associated AMPs are proved to participate in not only reproductive tract host defense but also other biological events such as sperm maturation [26,27] and sperm motility [28,29]. Although the AMPs identified in the reproductive system of marine animals have been widely studied for their antimicrobial activities, less is known about their potential functions in the reproductive process. SCY2 is a novel AMP identified from *S. paramamosain* in our previous study [30]. It is male specific and dominantly expressed in the ejaculatory duct (ED). During mating, SCY2 showed cross-gender transmission and was thought to exert reproductive immune functions [30]. Interestingly, the expression level of SCY2 is found significantly induced by PG, but not by lipopolysaccharides (LPS) [30]. Those previous results led us to presume whether SCY2 may not only exert antimicrobial activity but also have multiple functions in reproductive processes, especially in hormone-modulated post-mating events. Meanwhile, a novel SCY2-interacting protein, scyreprocin (MH488960), was revealed and proved to exert potent, broad-spectrum antibacterial, antifungal, and antibiofilm activities in vitro by multiple action modes [31]. Unexpectedly, recombinant products of SCY2 and scyreprocin showed no synergistic antimicrobial activity in vitro [31]. Considering both AMPs were dominantly expressed in the male reproductive system, whether their synergistic functions were reflected in other aspects of the reproduction processes as reported in other animals attracted us to explore their potential physiological roles in the present study.

In this work, the in vivo expression profiles of scyreprocin and SCY2 were investigated to confirm their roles in the reproductive immunity of mud crab. Interestingly, we found that both scyreprocin and SCY2 were also expressed in sperm, and their localizations were dynamically changed during the AR. To better understand sperm AR of mud crab, we used flow cytometry and Ca^2+^ fluorescent probes to investigate whether PG was one of the ARIS of crab sperm and to assess the change in {Ca^2+^}_i_ during AR. The potential functions of scyreprocin and SCY2 in AR were further explored by antibody blockade assays. In addition, the binding properties of scyreprocin and SCY2 to PG and Ca^2+^ were evaluated via functional experiments to further verify their roles in PG-induced AR.

## 2. Results

### 2.1. Expression Pattern of Scyreprocin and SCY2 In Vivo

Under natural conditions, the scyreprocin transcript was predominantly expressed in male gonads, with the highest expression level in testes, followed by anterior vas deferens, while relatively low expression was observed in female crabs, with the highest expression in ovaries (Figure 1A). High levels of scyreprocin protein expression were detected in gonads of adult male crabs, while no scyreprocin expression was detected in adult female crabs. (Figure 1B). In juvenile male crabs, scyreprocin was mainly expressed in spermatophores isolated from testes and seminal plasma isolated from ED (Figure 1C). In adult males, scyreprocin was detected in the spermatophore, and seminal plasma was collected from anterior vas deferens, seminal vesicle, ED, and posterior ejaculatory duct (Figure 1C). Detection of in situ expression indicated that scyreprocin was mainly expressed in spermatophores and epithelial cells of the testis, while SCY2 was expressed in interspaces between spermatophores in the testes of adult mud crabs (Figure 1D). Only weak signals of both proteins were detected in testicular sections of juvenile males (Figure 1D). Scyreprocin and SCY2 were not detected in spermathecae of pre-mating females. In post-mating females, scyreprocin and SCY2 signals were detected in the contents of spermathecae, moreover, strong SCY2 fluorescent signals were observed in epithelial cells (Figure 1E and Appendix A).

### 2.2. Scyreprocin and SCY2 Responded to Bacterial Infection In Vivo

A primary testicular cell culture method was established in this study (Appendix A). Microbial growth occurred during the first two days in several samples of cultured testicular cells. The endogenous microbes were isolated and identified as *Pseudomonas putida* (Appendix A), which was an aquatic pathogen. The isolated endogenous bacteria *P. putida* isolate X1 was susceptible to recombinant SCY2 (rSCY2) and recombinant Scyreprocin (rScyreprocin) treatments (Table 1). Scanning electron microscopy (SEM) observation revealed significant morphological changes in the bacterial membrane induced by rSCY2 and rScyreprocin treatments (Figure 2A). After being challenged with the *P. putida* isolate X1, in vitro cultured spermatophores showed a significant increase in scyreprocin and SCY2 expression (Figure 2B). Expression levels of SCY2 and scyreprocin were induced in testes and EDs after in vivo challenge with *P. putida* isolate X1 (Figure 2C,D). These results showed that scyreprocin and SCY2 could effectively inhibit and kill the pathogenic bacteria of mud crab and had a positive in vivo response to pathogen infection, indicating their roles in reproductive immunity.

### 2.3. Subcellular Localization of Scyreprocin and SCY2 in Mud Crab Sperm

Spermatids at various spermiogenesis stages were observed in cultured testicular cells (Figure 3A). In the early proacrosomal granule phase, SCY2 and scyreprocin were co-localized in the cytoplasm. In the preacrosomal vesicle phase, the nucleus shape started to change and preacrosomal granules (PGs) aggregated to form a proacrosomal vesicle (PV). SCY2 and scyreprocin signals were detected on PGs and PV, but rarely co-localized. In the preacrosomal phase, the nucleus developed into a cup-like shape and enwrapped the PV. SCY2 was expressed on the outer edge of the PV, while scyreprocin was located in unaggregated PGs (Figure 3A). The two later spermiogenesis stages (acrosome phase and mature phase) were not observed in the in vitro cultured testicular cells in the present study.

In mature sperm, organelle staining assays revealed that scyreprocin and SCY2 were co-localized in the endoplasmic reticulum (ER), Golgi apparatus, and mitochondria (Figure 3B). Transmission electron microscopy (TEM) observations yielded a refined image of scyreprocin and SCY2 localization in single sperm, where they showed co-localization in mitochondria, central tube, and apical cap (AC) (Figure 3C).

### 2.4. Progesterone Induced In Vitro Sperm Acrosome Reaction of S. paramamosain

To investigate the potential roles of SCY2 and scyreprocin in sperm AR, sperm collected from male gonad and female spermathecae were used for in vitro AR-induction tests (Figure 4A). When treated with artificial seawater containing 0.3% (*w*/*v*) Ca^2+^ (ASW), the AR ratio (%AR) increased significantly in sperm collected from spermathecae, whereas those collected from males showed no statistical difference compared to the control group (Figure 4B,C). These results suggested that some components in spermathecae might be requisite for sperm AR.

In a year-long investigation on SCY2 expression in male crabs, the highest transcriptional level of SCY2 was observed during mating seasons (May–July, October–December) (Appendix A) and the change in the SCY2 expression level is consistent with that of the hormones [30], indicating the correlation between PG and SCY2 expression levels. In post-mating female crabs, the PG level in the ovary increased during oogenesis, with the highest PG level occurring near ovarian maturation (pre-ovulation) (Figure 4E). Therefore, we further investigated whether PG could induce AR. Compared with the control group (mean ± SD = 22.76 ± 7.82%), the %AR of sperm collected from males was significantly induced by ASW containing PG (38.92 ± 1.22%) (Figure 4B,C), and sperm at different AR stages were observed by SEM (Figure 4D). The in vivo assay showed that sperm {Ca^2+^}_i_ increased significantly when males were directly injected with PG (Appendix A). These results indicated that PG could induce the AR of mud crab sperm, and {Ca^2+^}_i_ could be used as an indicator for mud crab sperm AR.

### 2.5. Localizations of SCY2 and Scyreprocin in Sperm during Sperm Acrosome Reaction

Sperm collected from spermathecae were treated with ASW to induce AR. Subcellular locations of SCY2 and scyreprocin were revealed by cellular immunofluorescence and immuno-colloidal gold technique. At AC protruding stage, SCY2 was detected in the AC, and scyreprocin was found in the cytoplasm. Scyreprocin was detected in the AV at the AV valgus stage and all the subsequent AR stages (Figure 5A). TEM observation showed that scyreprocin could be detected not only in AV but also in the acrosomal vesicle membrane and mitochondria (Figure 5B).

### 2.6. SCY2 and Scyreprocin Participated in Progesterone-Induced Acrosome Reaction

To investigate the possible roles of SCY2 and scyreprocin in sperm AR, sperm (collected from males) were incubated with Ca^2+^-free ASW (Ca^2+^-FASW) containing SCY2 and/or scyreprocin antibodies and then treated with PG to induce AR. Samples were analyzed for the %AR and {Ca^2+^}_i_ (Figure 6A).

In the antibody control groups, sperm co-incubated with scyreprocin antibody showed no change in {Ca^2+^}_i_, while sperm treated with the SCY2 antibody showed a significant decrease in {Ca^2+^}_i_ (Figure 6B–D). Hence, scyreprocin and SCY2 played different roles in maintaining intracellular Ca^2+^ homeostasis. 

Progesterone significantly induced sperm AR (66.27 ± 8.88% reacted) after 22 h in comparison with the untreated control (23.00 ± 0.32% reacted) (Figure 6B,C), but it could not induce the AR of the sperm pretreated with either SCY2 antibody (8.13 ± 0.20% reacted) or scyreprocin antibody (27.64 ± 0.75% reacted) (Figure 6B,C). After replenishing recombinant proteins to the corresponding antibody-treated samples, both %AR and {Ca^2+^}_i_ were restored (Figure 6B–D). Although antibody blockade of scyreprocin did not affect the {Ca^2+^}_i_ of unreacted sperm, it did hinder PG from inducing sperm AR (Figure 6B–D). These results indicated that scyreprocin might act as an important mediator in initiating PG-induced AR. Similarly, PG could not trigger sperm AR when scyreprocin and SCY2 were inhibited simultaneously (6.34 ± 0.28% reacted). Later replenishment of both rScyreprocin and rSCY2 allowed the increase in %AR (42.66 ± 2.96% reacted) and {Ca^2+^}_i_ (Figure 6B–D). Besides, SCY1, the SCY2 homologous protein, showed different localization from SCY2 in sperm (Appendix A) and had no detectable effect on the PG-induced {Ca^2+^}_i_ increase (Appendix A).

### 2.7. Progesterone Binding Capacity of SCY2 and Scyreprocin

Progesterone has been shown to induce SCY2 expression in our previous study [30]. We performed a modified ELISA assay to further determine the interaction of PG with rSCY2 and rScyreprocin, respectively. Scatchard plot analysis showed that the PG-binding affinity of rScyreprocin/rSCY2 mixture (calculated equilibrium dissociation constant, K_D_ = 72.2 nM) was stronger than that of rScyreprocin (K_D_ = 258.9 nM) and rSCY2 (K_D_ = 143.0 nM) alone, thus indicating that the PG-binding affinity was enhanced in the presence of both proteins.

### 2.8. SCY2 was Involved in Calcium Influx during Acrosome Reaction

Antibody blockade of SCY2 resulted in a significant decrease in sperm {Ca^2+^}_i_ (Figure 6D). To investigate its possible functions, sperm were treated with the SCY2 antibody or Ni^2+^ (set up as a Ca^2+^ channel inhibition control group). The samples were analyzed for the {Ca^2+^}_i_ (Figure 7C) and %AR (Figure 7D,E). 

After SCY2 antibody treatment, the sperm {Ca^2+^}_i_ was markedly reduced to a level similar to that of the positive control (Ni^2+^-treated group), and the Ca^2+^ influx induced by PG treatment was inhibited, suggesting the association between SCY2 and Ca^2+^ influx (Figure 7C). Similarly, flow cytometry analysis showed that treatments of Ni^2+^ and SCY2 antibody inhibited %AR of the sperm samples (Figure 7D,E). In the Ca^2+^-dependent gel-shifting assay, an overt band-shift of rSCY2 was observed in the presence of CaCl_2,_ which was more distinct in the presence of both ethylene glycol tetraacetic acid (EGTA) and CaCl_2_ (Figure 7F). It is worth noting that SCY2 showed no binding affinity to Mg^2+^ (Figure 7F).

## 3. Discussion

The mating behavior of marine animals provides an opportunity for pathogens in the aquatic environment to infect sperm and enter the female reproductive system. Crab sperm experience long-term storage (weeks to months) in the spermathecae before insemination [32], thus, bioactive molecules such as AMPs in the reproductive system are requisite for sperm health. As AMPs are highly expressed in male gonads, scyreprocin and SCY2 could efficiently inhibit the growth of aquatic pathogens such as the isolated *P. putida* strain X1 (Table 1). Both AMPs could be transferred to female spermathecae during mating and maintained in spermathecae until ovulation (Appendix A), thus it would be inferred that they may provide prolonged protection for the reproduction process of mud crab. Unlike other AMPs identified in mud crabs [31,33,34], rSCY2 only had moderate antimicrobial activity in vitro [30], and unexpectedly, showed no in vitro synergistic antimicrobial activity with rScyreprocin [31]. These findings prompt us to explore whether the interaction between SCY2 and Scyreprocin is reflected in other reproductive processes beyond reproductive immunity.

With numerous reproductive-associated AMPs being successively identified in the past decades, the fact that the male-specific AMPs, such as SCY2 and scyreprocin [30], are often highly expressed in the genital system during breeding seasons seems to raise a contradiction against reproduction-immunity trade-offs [35]. In recent years, AMPs have been shown to function beyond their antimicrobial activity during reproduction. The potential dual roles of β-defensins in the regulation of infection and control of sperm function is compelling [36]. Rat epididymis-specific β-defensin 15 plays a dual role in both sperm maturation and pathogen defense in rat epididymis [29]. Rat Bin1b is proved important for the acquisition of sperm motility and the initiation of sperm maturation [27]. Moreover, human cathelicidin 18 in seminal plasma is processed to generate a 38-amino acid AMP (ALL-38), transferred to the female reproductive tract, and enzymatically activated upon exposure to the vaginal milieu, preventing infection following sexual intercourse [37]. SCY2 shares similar cross-gender transmission patterns with ALL38 and its expression is regulated by PG but not LPS [30]. Therefore, we hypothesized that SCY2 may play an unrevealed role in hormone-regulated post-mating events in addition to its antimicrobial activity. In this study, we found that scyreprocin and SCY2 were presented not only in seminal plasma but also on the sperm apical cap (Figure 3) and were detected in the sperm apical cap (both AMPs) and acrosomal vesicle (scyreprocin) during PG-induced AR (Figure 5). These findings strongly support our prior presumption. We have confirmed that scyreprocin and SCY2 existing in seminal plasma provide anti-infection protection (Figure 2), but what are the potential biological functions of the AMPs located on sperm during AR? 

Acrosome reaction, one of the hormone-associated sperm activities, is finely regulated by hormones, ion channels, and preassemble signal pathways [5,20,38,39]. It is now well known that in mammals and sea urchins, the sperm AR requires the collaboration of hormones (e.g., PG) and cationic channels (e.g., CatSper). In most species, egg water containing a variety of reproduction-related hormones is considered the main contributor to sperm activities [40]. PG is considered an important factor to endow sperm fecundity by initiating sperm capacitation (motility), hyperactivation, and AR [41,42]. Crab sperm is non-flagellated and thus lacks motility [25]. Researchers have tried to induce the AR of crab sperm isolated from the male reproductive system by means of ionic carriers and extracted infraspecific egg-water, but results were inconsistent among different crab species [24,43,44,45,46,47]. Therefore, it is still controversial as to what triggers the AR of crab sperm. In the present study, sperm isolated from spermathecae but not the sperm isolated from the male reproductive system could directly undergo AR after ASW treatment (Figure 4B,C), suggesting certain components in spermathecae were requisite for sperm AR. Consistent with previous reports [48], this study showed that the PG level of the female ovary gradually increased after mating and peaked before ovulation (Figure 4E). After PG treatment, sperm collected from males showed a significant increase in %AR (Figure 4B,C), suggesting that PG may be one of the ARIS for mud crab sperm.

PG induces AR through its membrane receptors on human sperm [1]. Screening of PG membrane receptors has been the focus of the study on the PG non-genomic effects. Some PG-binding proteins have been identified on the sperm membrane, among which CatSper and PAQR7 are thought to be the most promising candidates as PG receptors [18,49]. Since the AR-associated molecules (e.g., CatSper) have not been identified in mud crab, the molecular basis of PG-induced AR of mud crab remains unclarified. Although some AMPs have been proven to play vital roles in various sperm activities (e.g., sperm maturation and motility) [27,28,29], to our knowledge, there are no reports that they are directly involved in sperm AR in any animal species. Based on our findings, questions were then raised: is there a close relationship between SCY2, scyreprocin, PG, and PG-induced AR? Do SCY2 and scyreprocin exert a similar role in crab sperm AR as the AR-associated components in other species?

In this study, we found that antibody blockade of either scyreprocin or SCY2 led to failure in PG-induced AR and {Ca^2+^}_i_ increase. Notably, antibody blockade of SCY2, rather than scyreprocin, led to a significant decrease in sperm {Ca^2+^}_i_ before PG treatment at levels similar to those in the Ni^2+^-treated group (Figure 7C). Later functional studies revealed the Ca^2+^-binding capacity of rSCY2 (Figure 7F). These findings strongly implied that SCY2 was important for the maintenance of Ca^2+^ homeostasis in mud crab sperm and participated in the Ca^2+^ influx during AR. Based on these results, it could be inferred that SCY2 may be involved in the active Ca^2+^ transportation and is a key component of the Ca^2+^ channel in mud crab sperm. Thus, further genomic screening of scygonadin homologous proteins and structural analysis of SCY2 may shed light on its basis for Ca^2+^ selectivity and its actual role in Ca^2+^ regulation.

Although antibody blockade of scyreprocin had no effect on {Ca^2+^}_i_ before PG induction, it could not induce AR in the absence of scyreprocin (Figure 6D). It was inferred that scyreprocin may be involved in the process of sperm receiving progesterone signals. Later functional studies confirmed the PG binding capacity of rScyreprocin and rSCY2 (Figure 7A), indicating that PG could bind to scyreprocin on the sperm and subsequently initiate the SCY2-mediated Ca^2+^ influx. Previous research has shown that the calcium channel CatSper is also a non-genomic PG receptor of human sperm [50]. SCY2 and scyreprocin were a pair of interacting proteins. Whether the complexes they formed play a similar role in sperm AR as CatSper is worth further exploration. 

Elevation of {Ca^2+^}_i_ during AR is caused not only by the influx of extracellular Ca^2+^ but also by the following Ca^2+^-induced Ca^2+^ release (CICR); that is, the Ca^2+^ would further induce the release of Ca^2+^ from intracellular Ca^2+^ stores such as acrosome and mitochondria [17]. In the present study, we did not permeabilize the sperm before antibody treatments; therefore, only scyreprocin and SCY2 on the cell surface were inhibited. Scyreprocin and SCY2 on sperm AC assumed crucial roles in the initial influx of Ca^2+^. Given that these two AMPs were also co-located with organelles in crab sperm (Figure 2), it remains to be investigated whether they would take part in the following CICR process.

During sperm AR, scyreprocin was seen in the AV (Figure 5A). The acrosomal vesicle is known to contain a variety of enzymes that dissolve the oolemma and assist in successful sperm–egg fusion [51]. Our preliminary experiments indicated that rScyreprocin exerted acid phosphatase and superoxide dismutase activity (54.84 U gprot^−1^ and 118.12 U mgprot^−1^, respectively). More studies are required to verify if scyreprocin in the AV exerts a similar function. During the mating season, male crabs expressed a synchronous increase in SCY2 in parallel with elevated PG levels [30]. Similarly, a significant increase in the SCY2 fluorescent signal was detected in the spermathecal epithelium after mating (Figure 1E). In a prior study on spermatophore transplantation, it was found that spermatophores could be absorbed into the spermathecal epithelium by endocytosis, and thus some spermatophore degradation products could enter the vessel lumen and further modulate female reproductive behavior [52]. It remains to be determined whether SCY2 detected in the spermathecal epithelium was a result of spermatophore degradation absorption or in situ expression. The origin and other possible physiological functions of SCY2 in the spermathecal epithelium thus also need further investigation.

## 4. Materials and Methods

### 4.1. Animals

Mud crab (*S. paramamosain*) were obtained from the Xiamen aquatic products market, Fujian, China. Crabs were acclimated in cement tanks containing seawater for 1–2 days before experiments. Before sampling, mud crabs were anesthetized by ice-bathing for 15 min and all efforts were made to minimize suffering. All animal experiments were carried out in strict accordance with the National Institute of Health Guidelines for the Care and Use of Laboratory Animals and were approved by the Animal Welfare and Ethics Committee of Xiamen University. 

### 4.2. Isolation of Spermatophores and Seminal Plasma

Semen was divided into seminal plasma and spermatophores following the prior descriptions [53]. Spermatophores were treated with 0.25% trypsin (prepared in Ca^2+^-FASW) to obtain single sperm. Seminal plasma was stored at −20 °C and sperm were stored at 4 °C before use.

### 4.3. Preparation of Recombinant Proteins and Polycolonal Antibodies

Recombinant His-tagged scyreprocin (rScyreprocin) and SCY2 (rSCY2) were generated following prior descriptions [31,54]. Purified proteins were dialyzed, concentrated, and stored in 50 mM phosphate buffer (PB, pH 8.0) at −80 °C. Protein concentration was determined by the Bradford assay [55]. The scyreprocin antibody and SCY2 antibody were prepared as previously described [30,31].

### 4.4. Quantitative PCR

To investigate the expression profile of scyreprocin transcripts, tissues from three adult males and three adult females (300 ± 20 g in weight) were sampled. Tissues were flash-frozen in liquid nitrogen and stored at −80 °C. Total RNA and protein of each tissue (30 mg) were extracted by the Tripure reagent (Roche, Mannheim, Germany) following the manufacturer’s instruction. Real-time quantitative PCR (RT-qPCR) was performed on a Roto-Gene Q platform (QIAGEN, Hilden, Germany) using the SYBR Green assay (Roche, Mannheim, Germany). The primer sequences are listed in Appendix A. Absolute qPCR was carried out to evaluate the expression profile of scyreprocin in different tissues of healthy adult male and female crabs (*n* = 3) as previously described [56]. For relative qPCR, the target gene Scyreprocin (GenBank Accession No. MH488960) was detected, and the Sp-β-actin (GenBank Accession No. GU992421) was chosen as the reference gene. Data were analyzed using the algorithm of the 2^−ΔΔCt^ method [57].

### 4.5. Antimicrobial Assays

The antimicrobial activity of the rScyreprocin, rSCY2, and rSCY2/rScyreprocin isomolar concentration mixture (e.g., the 1 μM mixture was composed of 1 μM rSCY2 and 1 μM rScyreprocin) against the isolated endogenous bacteria was evaluated. The MIC and MBC values were determined in triplicate on separate occasions following the prior descriptions [54]. After 30 min-treatment with rScyreprocin (2 μM) or rSCY2 (4 μM), the morphological changes of the isolated endogenous bacteria were observed using a Zeiss Supra™ 55 Scanning Electron Microscope (Carl Zeiss Microscopy GmbH, Oberkochen, Germany) as described earlier [31,58]. Experiments were performed three times on different occasions. 

### 4.6. Western Blotting

To study the expression profile of scyreprocin, total proteins extracted from the tissues of adult male and female crabs (*n* = 3) were submitted to the Tricine-SDS-PAGE assay and transferred to a polyvinylidene difluoride (PVDF) membrane (Amersham, Sunnyvale, CA, USA). Immune detection of Sp-β-actin and scyreprocin was carried out using β-actin antibody (Santa Cruz Biotechnology, Santa Cruz, CA, USA) and scyreprocin antibody (dilution factors = 1:1000) following the standard Western blotting procedures. To evaluate expression level of scyreprocin in semen, seminal plasma and spermatophores were isolated from adult (300 ± 10 g) and juvenile male crabs (100 ± 10 g). Total protein extracted from spermatophores and 5 μg seminal plasma protein (*n* = 3) were detected for scyreprocin expression as described above. To investigate the expression profile of scyreprocin and SCY2 after bacterial infection, *P. putida* isolate X1 was injected into adult male crabs (250 ± 10 g) at 3 × 10^3^ cfu crab^−1^. Crab saline injections were performed as controls (*n* = 3). After 24 h, testes and ED samples (*n* = 3) were subjected for scyreprocin and SCY2 detection, and blots were quantified and analyzed using ImageJ (National Institutes of Health, Bethesda, MD, USA).

### 4.7. Immunofluorescence Assay

To explore the in situ expression profile of scyreprocin and SCY2, testes of juvenile (50 ± 10 g) and adult males (300 ± 10 g), spermathecae of female crabs on the day after mating, post-mating stage I, post-mating stage II, post-mating stage III, pre-ovulation stage were collected and sectioned (8–10 μm) for immunofluorescence assay. To investigate the in situ expression of scyreprocin and SCY2 in male gametes, in vitro-cultured testicular cells (seeded at 2 × 10^6^ cells well^−1^ for 3 days, Appendix A) and sperm isolated from male crabs (250 ± 10 g) were fixed with 4% paraformaldehyde (prepared in crab saline), permeabilized with 0.1% Triton X-100. To study the expression of scyreprocin and SCY2 after bacterial infection, the in vitro cultured spermatophore were incubated with *P. putida* isolate X1 (100 cfu well^−1^) for 24 h before being subjected to immunofluorescence assay. Immunofluorescence assay was carried out based on a prior description [30]. Briefly, samples were blocked with 5% bovine serum albumin (BSA), incubated with a mixture containing SCY2 and scyreprocin antibodies (1:400) or unimmunized serum (1:100) for 4 h at 37 °C in a humidified chamber. After washing with phosphate-buffered saline (PBS, pH 7.4), samples were incubated with Dylight 488 conjugated goat anti-mouse IgG and Dylight 650 conjugated goat anti-rabbit IgG secondary antibodies (1:1000) (Thermo Fisher Scientific, Waltham, MA, USA) for 1 h. Slides were mounted with coverslips using Vectashield^®^ antifade mounting medium with DAPI (Vector Labs, Burlingame, CA, USA), and observed by a confocal laser scanning microscope (Zeiss Lsm 780 NLO; Carl Zeiss, Jena, Germany).

### 4.8. Hormone Level Examination

In every month of the year 2012, EDs from 3 crabs (300 ± 20 g) were tested for PG, testosterone, and estradiol levels using ELISA kits (Cayman Chemical Company, Ann Arbor, MI, USA). Tissues (~50 mg) were ground in liquid nitrogen and mixed with 1 mL of ELISA buffer (Cayman Chemical Company, Ann Arbor, MI, USA); supernatants were subjected to ELISA following the manufacturer’s instructions. Analyses were carried out in duplicate. For evaluation of PG level in pre- and post-mating females, spermathecae and ovaries from un-mated females, female crabs at the day after mating, post-mating stage I, II, III, pre-ovulation, and post-ovulation stage, were collected (*n* = 6). The samples (~30 mg) were analyzed as described above. 

### 4.9. Enzyme-Linked Immunosorbent Assay

To test if rSCY2, rScyreprocin, and/or rSCY2/rScyreprocin could bind to PG (Sigma-Aldrich, St. Louis, MO, USA), a modified ELISA assay was performed following a prior description [31]. Briefly, a flat bottom 96-well ELISA plate was coated with PG (3 μg), blocked with 5% BSA, and incubated with serial dilutions of rScyreprocin, rSCY2, and rSCY2/rScyreprocin (0–24 μM, 100 μL well^−1^). After washing with PBS (50 mM, pH 7.4), plates were incubated with 100 μL mixture containing scyreprocin antibody (1:2000) and SCY2 antibody (1:1000) for 2 h before incubating with a mixture of HRP-labeled goat anti-rabbit IgG and HRP-labeled goat anti-mouse IgG (1:5000) for 1 h. After the colorimetric reaction, absorbance at 450 nm was measured using a multifunctional microplate reader (TECAN GENios; Tecan Group Ltd., Männedorf, Switzerland). The assays were carried out in triplicate and the results were analyzed using Scatchard plot analysis.

### 4.10. Transmission Electron Microscopy (TEM) Observation

Sperm freshly collected from mature male crabs (250 ± 10 g) were fixed in 4% paraformaldehyde. For TEM observation, samples were subjected to ultrathin sections and negative stained following standard protocols before being observed by a transmission electron microscope (FEI, Tecnai G2 F20, Eindhoven, the Netherlands) [59]. For the immunocolloidal gold assay, ultrathin sections were blocked with 5% BSA (prepared in PBS, pH 7.4) for 30 min and incubated overnight with a mixture of scyreprocin antibody (1:100) and SCY2 antibody (1:50) at 4 °C. The scyreprocin antibody was recognized by the specific secondary antibody coupled with 6 nm of colloidal gold (Electron Microscopy Sciences, Ft. Washington, PA, USA), and the SCY2 antibody was revealed with a 25 nm colloidal gold-coupled secondary antibody (Electron Microscopy Sciences, Ft. Washington, PA, USA). Sections were post-fixed with 4% paraformaldehyde, negative stained, and subjected for TEM observation.

### 4.11. SCY2-Calcium Binding Property

A modified electrophoretic mobility shift assay (EMSA) was used to investigate the Ca^2+^ binding property of SCY2. rSCY2 (2 μg) and rSCY2 incubated in Tris-HCl (10 mM, pH 7.5) containing EGTA (0.1 mM) at room temperature for 3 h, were set up as blank and experimental controls, respectively. Samples of (1) rSCY2 incubated in Tris-HCl containing CaCl_2_ (0.1 mM) for 3 h, (2) rSCY2 first incubated with CaCl_2_ (0.1 mM) for 3 h and then supplemented with EGTA (final concentration = 0.1 mM) for 10 min, and (3) an experimental group designed in reverse order were subjected to native gel electrophoresis. Similar assays using (1) Mg^2+^ and EDTA, and (2) Ni^2+^ and EDTA were carried out as described above.

### 4.12. Evaluation of Sperm Intracellular Calcium Concentration and Acrosome Reaction Ratio

Methods were developed in the present study to assess the sperm %AR and {Ca^2+^}_i_. A modified method based on flow cytometry was set up to analyze %AR. Sperm samples were analyzed using a CytoFLEX LX (Beckman Coulter Inc., Brea, CA, USA) and data were acquired with CytExpert software (Version 2.0). Samples were stained with DAPI (Thermo Fisher Scientific) and events with DAPI fluorescence were gated as intact sperm (valid counted events). The number of events was set to 8000 and recorded. The recorded sperm population was divided into two non-overlapping sub-populations representing acrosome-reacted and non-reacted sperm. For {Ca^2+^}_i_ evaluation, sperm samples were loaded with Fluo-4/AM following the manufacturer’s instruction, and fluorescence intensity was measured using a microplate reader (TECAN GENios; Tecan Group Ltd.).

### 4.13. In Vitro AR Induction

Sperm were freshly collected and randomly divided into aliquots. To investigate the difference in %AR between sperm samples (2 × 10^7^ sperm mL^−1^) collected from male gonads (*n* = 3) and female spermathecae (*n* = 3), (1) control groups (male- and female-derived sperm suspended in Ca^2+^-FASW for 24 h), (2) ASW groups (male- and female-derived sperm suspended in ASW for 24 h), and (3) PG-treatment groups (male-derived sperm suspended in ASW containing 20 μg mL^−1^ PG for 24 h) were analyzed for %AR and {Ca^2+^}_i_ as described before. Male-derived sperm after PG treatment were subjected for SEM observation following prior description [58].

### 4.14. Antibody Blockade Assay

Sperm were collected from male gonads, washed in Ca^2+^-FASW, randomly divided into aliquots, and used to study the functions of scyreprocin and SCY2 in the AR process. To explore the possible role of SCY2 in Ca^2+^ influx during PG-induced AR, Ni^2+^ (final concentration = 5 μM) was applied as a Ca^2+^ channel inhibitor. Sperm (from 3 crabs) were adjusted to 2 × 10^7^ sperm mL^−1^ and experimental groups were set up as follows: (1) non-treatment control group (sperm incubated in ASW for 24 h), (2) positive control group (sperm treated with Ni^2+^ for 2 h then incubated in ASW for 22 h), (3) positive PG-treated group (sperm treated with Ni^2+^ for 2 h then incubated in ASW containing 50 μg mL^−1^ PG for 22 h), (4) SCY2-blocked group (sperm treated with SCY2 antibody for 2 h then incubated in ASW containing 50 μg mL^−1^ PG for 22 h), and (5) SCY2-blocked PG-treated group (sperm treated with the SCY2 antibody for 2 h then incubated in ASW containing 50 μg mL^−1^ PG for 22 h). The dilution factor of the SCY2 antibody applied was 1:500 (Appendix A). Samples were analyzed for %AR and {Ca^2+^}_i_ as described previously. 

To investigate the involvement of scyreprocin and SCY2 during AR, sperm (~1 × 10^6^ cells mL^−1^ in Ca^2+^-FASW) were treated with scyreprocin antibody (1:1000), SCY2 antibody (1:500), and scyreprocin (1:1000)/SCY2 (1:500) antibody mixture for 2 h, respectively, before PG treatment (50 μg mL^−1^ in ASW) for 22 h. Optimization of antibody concentration was confirmed by a preliminary study (Appendix A). Samples were subjected to {Ca^2+^}_i_ and %AR elevation. To confirm the physiological function of SCY2 and scyreprocin, the corresponding protein (4 μM) was replenished at 1 h before %AR and {Ca^2+^}_i_ evaluation.

### 4.15. Statistical Analysis

Statistical analyses were performed using IBM SPSS statistics (Version 22; IBM Corp., Armonk, NY, USA) and GraphPad Prism software (version 5.01; GraphPad Software Inc., San Diego, CA, USA). One-way analysis of variance (ANOVA) followed by Tukey post-test were used to compare the scyreprocin expression profile in different tissues of *S. paramamosain* and the levels of %AR and {Ca^2+^}_i_ of sperm samples after different treatments. Two-way ANOVA followed by Bonferroni post-test was performed to analyze the changes in scyreprocin and SCY2 in vivo expression levels before and after bacterial infection. Significant levels were accepted at *p* < 0.05.

## 5. Conclusions

The present study revealed that two male gonadal AMPs play a dual role in both reproductive immunity and PG-induced AR of mud crab *S. paramamosain* (Figure 8). Adult male crabs expressed SCY2 and scyreprocin in sperm and seminal plasma. The AMPs exerted their antimicrobial activity to provide anti-infection protection during reproduction. In post-mating females, PG level increased, reaching a peak value before ovulation, and inducing sperm AR upon sperm–egg attachment. Scyreprocin and SCY2 expressed on sperm directly participated in PG-induced AR. During the process, PG bound to scyreprocin and then triggered SCY2-mediated Ca^2+^ influx. The increase in {Ca^2+^}_i_ led to the AR and ultimately sperm–egg fusion. It was worth noting that PG failed to induce AR when either scyreprocin or SCY2 function was absent. Although the detailed functions of scyreprocin and SCY2 in sperm AR remain to be elucidated, the observed dual effect of scyreprocin and SCY2 attest to the importance of the reproduction-associated AMPs in *S. paramamosain*. Thus, a particular protein may exert distinct functions alone and/or with the assistance of its interacting partners under different physiological stages and at different action sites. This reproductive strategy of mud crabs may have evolved over millions of years to cope with their complex habitat. At present, there is a very limited understanding of the mechanism for crab sperm action, and the results of this study indicate that it may differ from that in well-studied mammals, sea urchins, and amphibians (Figure 9). Due to technical obstacles, we are currently unable to perform direct functional verification by constructing scyreprocin- or SCY2-deficient crabs. However, the suggestion that AMPs, beyond their antimicrobial activity, may participate in post-mating sperm activation may shed new light on the intricate interplay between immunity and reproduction. Moreover, given that AMP expression is under endocrine regulation and controls the breeding process in mammals, birds, and invertebrates [60,61], the results of the present study will be relevant for future studies on how reproductive hormones control the tradeoffs between reproduction and immunity.

## Figures and Tables

**Figure 1 ijms-23-03373-f001:**
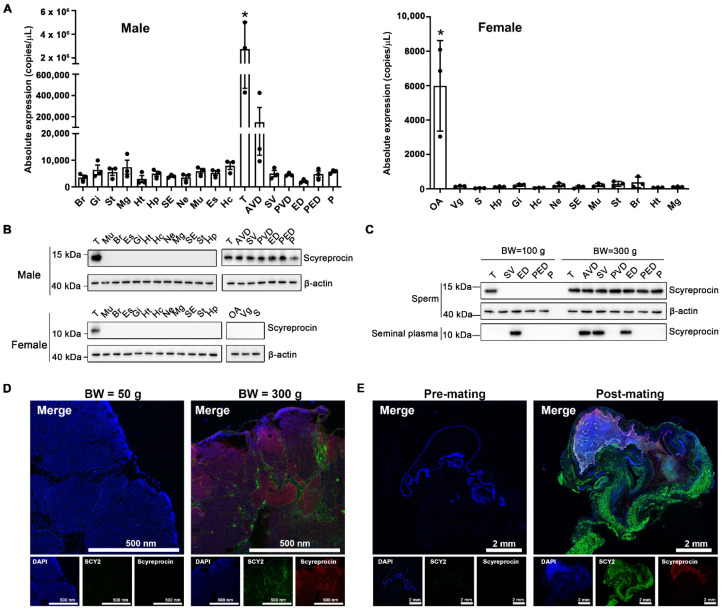
Scyreprocin and SCY2 expressed in reproductive system of adult male mud crabs and transferred to female spermathecae via mating. (**A**) Scyreprocin transcriptional expression level in adult male (*n* = 3) and female (*n* = 3) *Scylla paramamosain* under natural conditions. Data are presented as the mean ± standard deviation (SD). * *p* < 0.05, one-way analysis of variance (ANOVA) and Tukey post-test. (**B**) Scyreprocin expression profiles in different tissues of adult male and female crabs (*n* = 3). (**C**) Scyreprocin expression profiles in semen (sperm and seminal plasma) collected from adult and juvenile males (*n* = 3). BW, body weight. (**D**) In situ expression of SCY2 (green) and scyreprocin (red) in testes of juvenile and adult males. (**E**) In situ expression of SCY2 (green) and scyreprocin (red) in spermathecae of pre- and post-mating females. In panels (**D**,**E**), nucleus is shown in blue color. Abbreviations: Br, brain; Gi, gill; St, stomach; Mg, midgut; Ht, heart; Hp, hepatopancreas; SE, subcuticular epidermis; Ne, thoracic ganglion mass; Mu, muscle; Es, eyestalk; Hc, hemolymph cell; T, testis; AVD, anterior vas deferens; SV, seminal vesicle; PVD, posterior vas deferens; ED, ejaculatory duct; PED, posterior ejaculatory duct; P, penis; S, spermatheca; OA, ovary; Vg, vagina.

**Figure 2 ijms-23-03373-f002:**
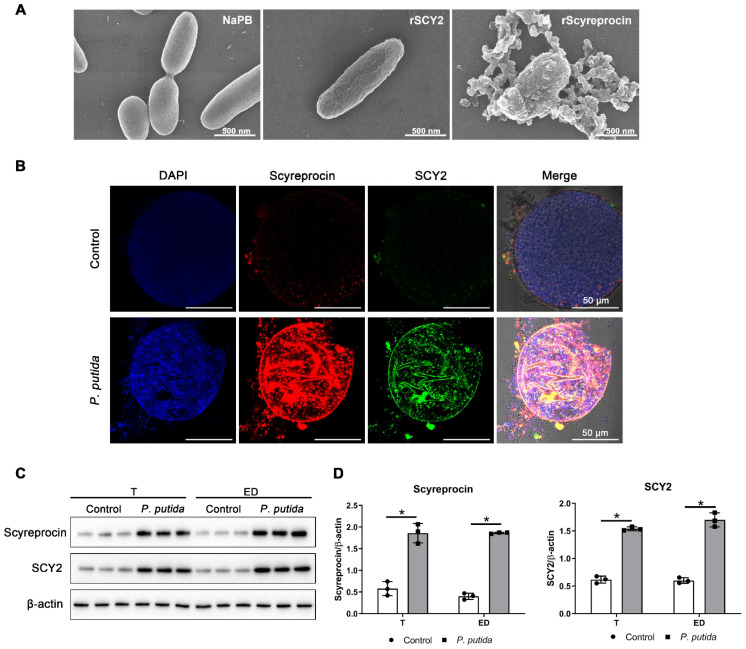
Scyreprocin and SCY2 responded to bacterial infections. (**A**) Morphological changes induced by recombinant scyreprocin (rScyreprocin) and SCY2 (rSCY2) in *Pseudomonas putida* isolate X1 (*n* = 3). *P. putida* isolate X1 (5 × 10^5^ cfu mL^−1^) was incubated with rScyreprocin (2 μM) or rSCY2 (4 μM) for 30 min and observed by a scanning electron microscopy. (**B**) Induction of SCY2 and scyreprocin expression levels in in vitro cultured spermatophores after *P. putida* isolate X1 challenge (*n* = 3). The in vitro cultured spermatophore were incubated with *P. putida* isolate X1 (100 cfu well^−1^) for 24 h before subjected to immunofluorescence assay. (**C**) Induction of SCY2 and scyreprocin expression levels in testis (T) and ejaculatory duct (ED) by in vivo *P. putida* isolate X1 challenge (*n* = 3). Adult male crabs were challenged with *P. putida* isolate X1 (3 × 10^3^ cfu crab^−1^). After 24 h, T and ED were sampled and subjected to Western blot analysis. (**D**) Quantification of the blots in (**C**) by ImageJ. Data are presented as the mean ± standard deviation (SD). * *p* < 0.05, two-way analysis of variance (ANOVA) and Bonferroni post-test.

**Figure 3 ijms-23-03373-f003:**
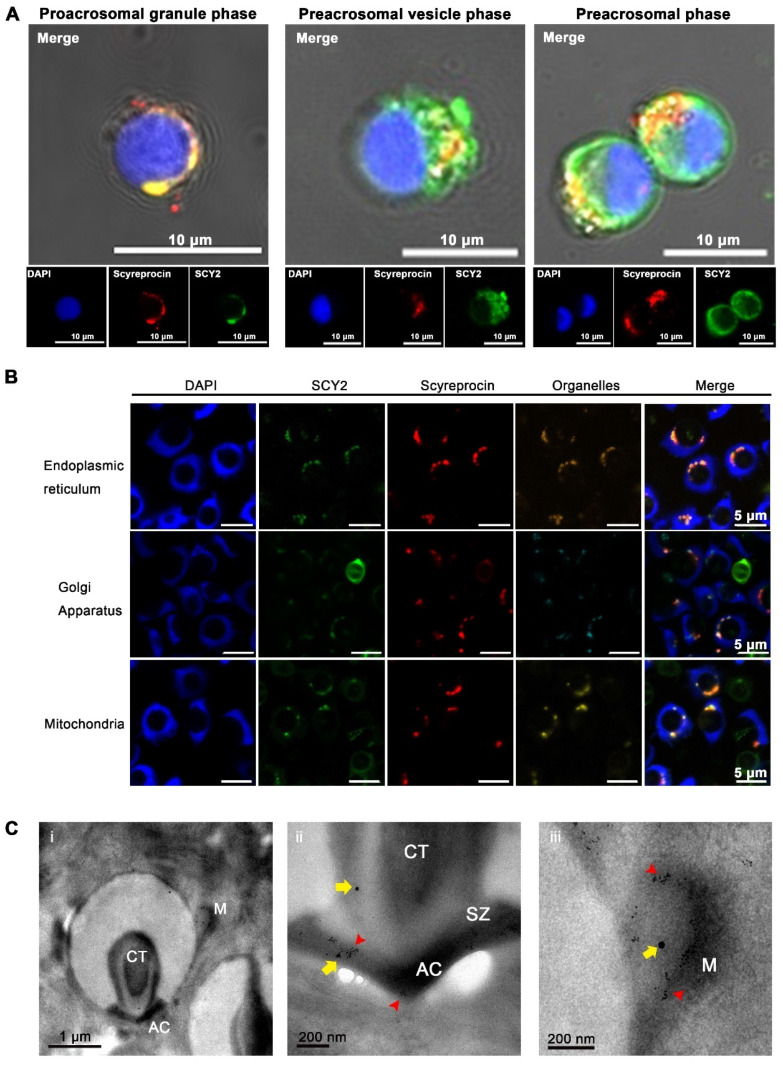
Subcellular location of scyreprocin and SCY2 in male gametes. (**A**) In situ expression of scyreprocin (red) and SCY2 (green) in spermatids at different spermiogenesis stages, nucleus was stained with DAPI (blue). In vitro cultured testicular cells (seeded at 2 × 10^6^ cells well^−1^ for 3 days) were subjected to immunofluorescence assay. (**B**) SCY2 and scyreprocin co-localized with organelles in sperm. Sperm were freshly isolated from seminal vesicles of adult male crabs and subjected to immunofluorescence assay. (**C**) In situ expression of scyreprocin and SCY2 observed by transmission electron microscope (TEM) in mud crab sperm: i, intact sperm; ii, apical cap (AC); iii, mitochondria (M). Red arrows: scyreprocin; yellow arrows, SCY2. Abbreviations: SZ, sub-cap zone; CT, central tube.

**Figure 4 ijms-23-03373-f004:**
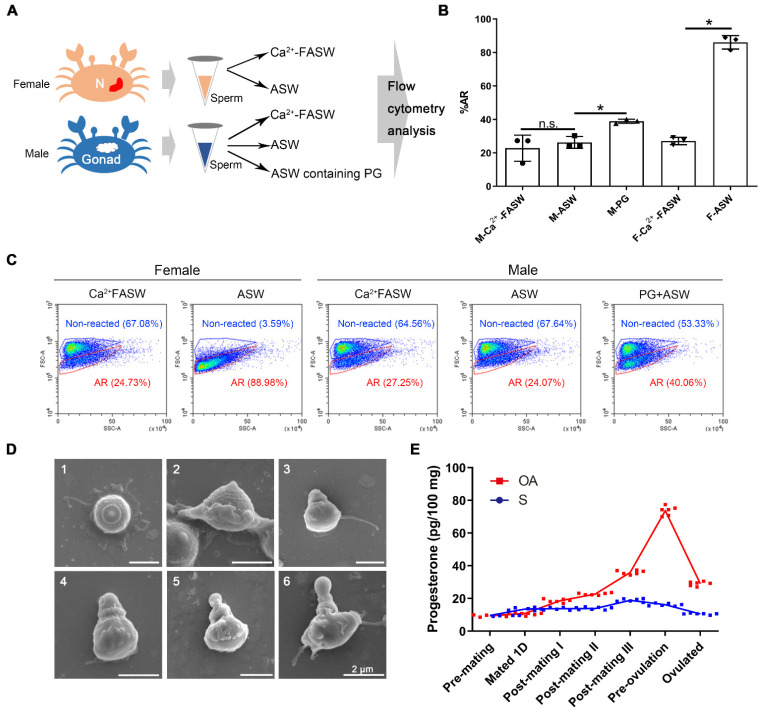
Progesterone (PG) was a crucial acrosome reaction (AR)-induced substance for crab sperm. (**A**) Schematic presentation of the AR ratio (%AR) evaluation on the sperm collected from male and female crabs. N, spermathecae; ASW, artificial seawater; Ca^2+^-FASW, Ca^2+^-free ASW. (**B**) Statistical analysis on %AR of sperm collected from female spermathecae and male gonads (*n* = 3). Data are presented as the mean ± standard deviation (SD). * *p* < 0.05, one-way analysis of variance (ANOVA) and Tukey post-test; n.s., not significant. M, male; F, female. (**C**) Flow cytometry assessment on %AR of sperm collected from female spermathecae and male gonads (*n* = 3). Sperm samples were treated with ASW, Ca^2+^-FASW (male- and female-derived sperm), or ASW containing 20 μg mL^−1^ PG (male-derived sperm) for 24 h before subjected to flow cytometry analysis. (**D**) Ultrastructural changes of crab sperm during AR observed by scanning electron microscope (SEM). Male-derived sperm after PG treatment in (**C**) were subjected for SEM observation: 1–2, unreacted sperm; 3, acrosome protruding stage; 4, acrosomal vesicle valgus stage; 5, central tube extension stage; 6, reacted sperm. (**E**) Changes in PG level in spermatheca (S) and ovary (OA) at pre- and post-mating stages. Spermathecae and ovaries from un-mated females, female crabs at the day after mating, post-mating stage I, II, III, pre-ovulation, and post-ovulation stage, were collected (*n* = 6). The samples (~30 mg) were subjected to PG level analysis. Data are presented as the mean ± SD.

**Figure 5 ijms-23-03373-f005:**
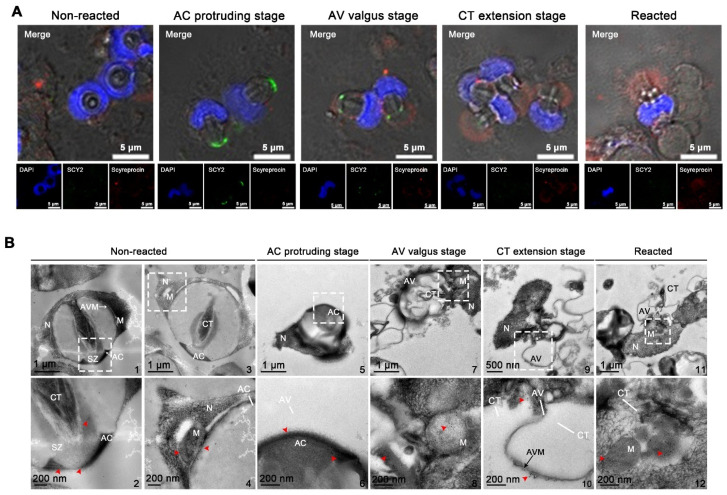
Expression pattern of scyreprocin and SCY2 in sperm during the acrosome reaction (AR). (**A**) Subcellular localization of scyreprocin and SCY2 in sperm at different AR stages (blue, nucleus; red, scyreprocin; green, SCY2). Male-derived sperm were treated with artificial seawater (ASW) containing 20 μg mL^−1^ PG for 24 h before subjected to immunofluorescence assay. (**B**) Subcellular localization of scyreprocin (red arrows) in sperm at different AR stages, from transmission electron microscopy (TEM) observation. Dashed lines indicate the zoom-in regions. Abbreviations: AC, apical cap; CT, central tube; M, mitochondria; SZ, sub-cap zone; AV, acrosomal vesicle; N, nucleus; AVM, acrosomal vesicle membrane.

**Figure 6 ijms-23-03373-f006:**
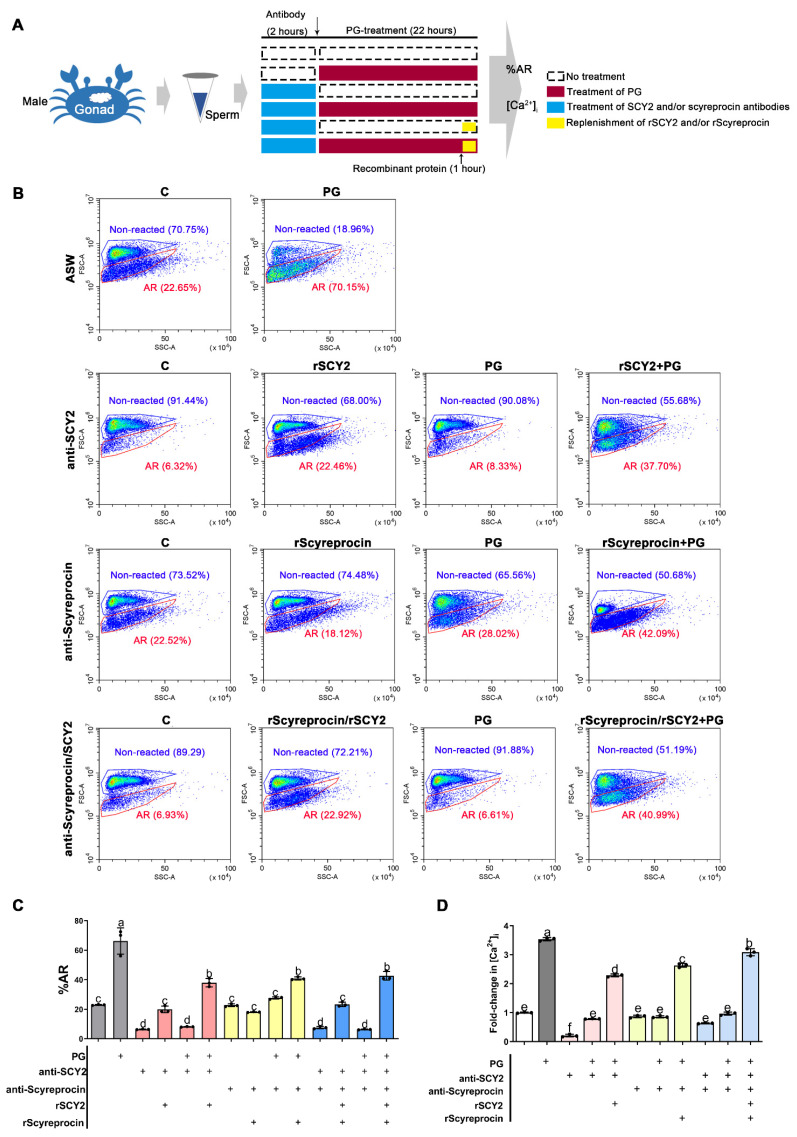
Scyreprocin and SCY2 functioned as critical molecules in progesterone (PG)-induced sperm acrosome reaction (AR). (**A**) Schematic presentation of the AR ratio (%AR) and intracellular Ca^2+^ concentration ({Ca^2+^}_i_) evaluation of the sperm collected from male and female crabs. (**B**) Flow cytometry analysis of the sperm %AR after different treatments. Male-derived sperm samples (~1 × 10^6^ cells mL^−1^) were pre-treated with SCY2 antibody (1:500) and/or scyreprocin antibody (1:1000) for 2 h and incubated with PG (50 μg mL^−1^ in artificial seawater) for 22 h. Samples were subjected to flow cytometry analysis. (**C**) Statistical analyses of the flow cytometry data presented in (**B**) (*n* = 3). Data are presented as the mean ± standard deviation (SD). (**D**) Evaluation of {Ca^2+^}_i_ in sperm samples in (**B**) (*n* = 3). In panels (**C**,**D**), “+” represents the addition of the corresponding component, data are presented as the mean ± SD. Letters denote significant differences, one-way analysis of variance (ANOVA), and Tukey post-test.

**Figure 7 ijms-23-03373-f007:**
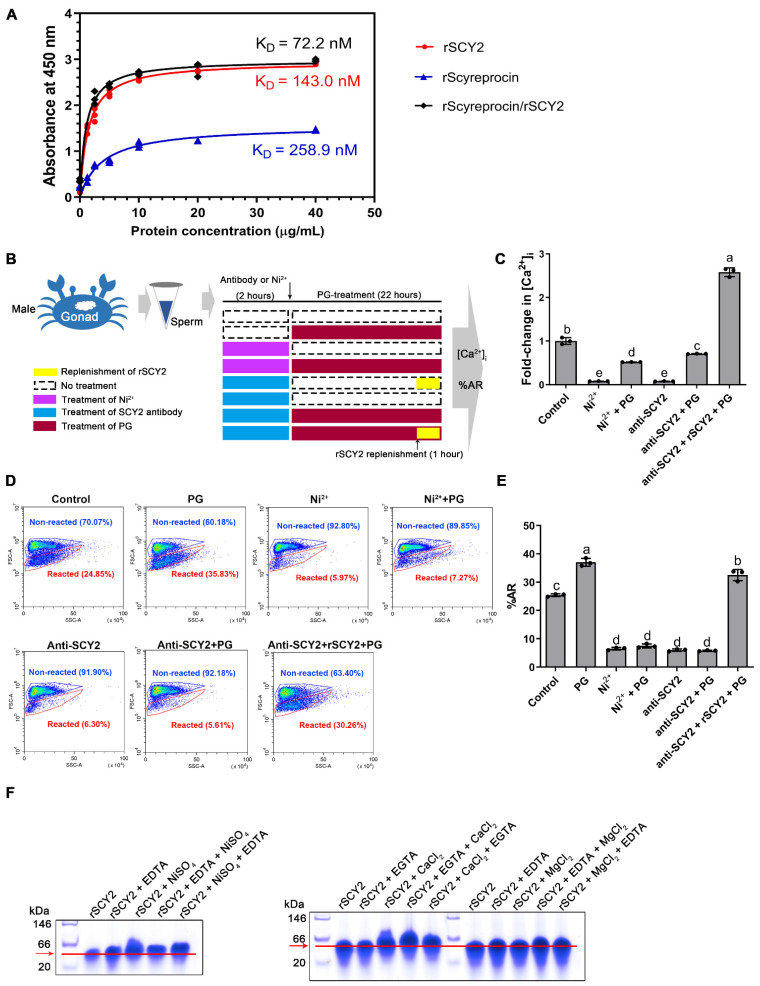
Interplay of scyreprocin, SCY2, progesterone (PG), and Ca^2+^. (**A**) Comparative examination of PG binding capacity of rScyreprocin, rSCY2 and rScyreprocin/rSCY2 mixture by a modified enzyme-linked immunosorbent assay (ELISA) (*n* = 3). (**B**) Schematic presentation of the intracellular Ca^2+^ concentration ({Ca^2+^}_i_) evaluation and acrosome reaction ratio (%AR) analysis. (**C**) Evaluation of {Ca^2+^}_i_ in the sperm pretreated with SCY2 antibody or Ni^2+^ (*n* = 3). Sperm (2 × 10^7^ sperm mL^−1^) were pre-treated with Ni^2+^ (5 μM) or SCY2 antibody (1:500) for 2 h, and incubated with PG (50 μg mL^−1^ in artificial seawater) for 22 h. Samples were subjected to {Ca^2+^}_i_ evaluation. (**D**) Flow cytometry analysis of the %AR (*n* = 3) of the samples in (**C**). (**E**) Statistical analysis of the data presented in (**D**). In panels C and E, data are presented as the mean ± SD. Letters denote significant differences, one-way ANOVA and Tukey post-test. (**F**) Electrophoretic mobility shift assays on the binding properties of rSCY2 with Ca^2+^, Mg^2+^, and Ni^2+^. rSCY2 (2 μg) was incubated in Tris-HCl containing CaCl_2_, MgCl_2_, or NiCl_2_ (0.1 mM) for 3 h and then supplemented with EGTA or EDTA (0.1 mM) for 10 min. Samples were subjected to native gel electrophoresis.

**Figure 8 ijms-23-03373-f008:**
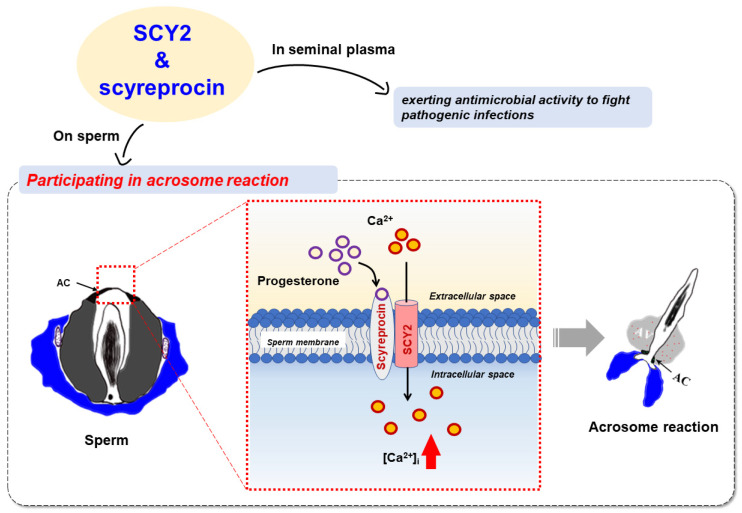
Roles of scyreprocin and SCY2 in the sperm acrosome reaction (AR) of *Scylla paramamosain*. Adult male crabs expressed scyreprocin and SCY2 in semen, which were then transferred to female spermatheca via mating. Scyreprocin and SCY2 in seminal plasma were proved to maintain gamete health by exerting antimicrobial activity. In sperm, scyreprocin and SCY2 showed co-localization on the apical cap and mitochondria, and are proven to participate in the initiation of progesterone-induced AR. In un-reacted sperm, SCY2 was responsible for maintaining intracellular Ca^2+^ homeostasis. Upon sperm–egg attachment, scyreprocin bound to progesterone, with SCY2 cooperatively strengthening the binding affinity. SCY2 bound to extracellular Ca^2+^ and transported it into the sperm. The increase in {Ca^2+^}_i_ ultimately initiated AR and allowed completion of sperm–egg fusion. Abbreviations: AC, apical cap; AV, acrosomal vesicle; {Ca^2+^}_i_, intracellular Ca^2+^ concentration.

**Figure 9 ijms-23-03373-f009:**
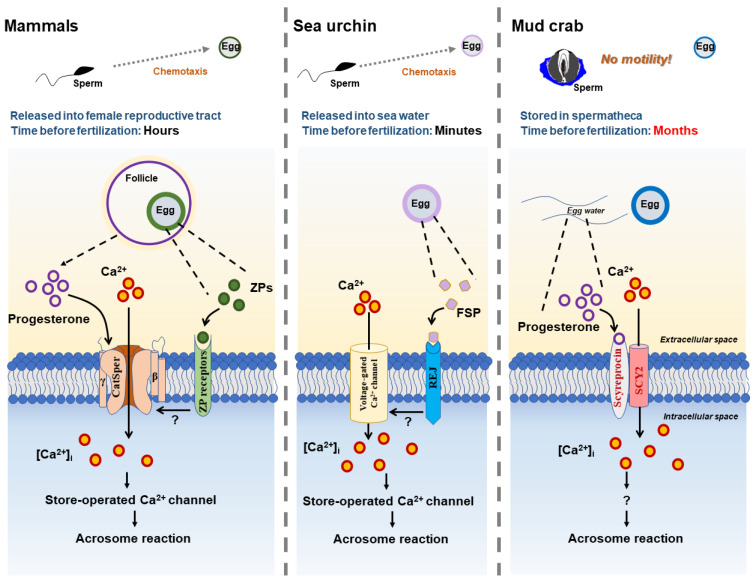
Brief comparison of sperm acrosome reaction (AR) mechanism in mammals, sea urchin, and mud crab. In mammals, progesterone and egg zona pellucida proteins (ZPs) interact with their corresponding receptors (i.e., CatSper, ZP receptors) on the sperm membrane, activate CatSper, and induce Ca^2+^ influx. Increase in intracellular Ca^2+^ concentration ({Ca^2+^}_i_) then leads to the release of Ca^2+^ from intracellular Ca^2+^ store, and thus completes sperm AR. In sea urchins, sperm AR is induced by the interaction of fucose sulfated glycoconjugate from egg-coat (FSG) and its specific receptor (REJ) on the sperm membrane, which opens a Ca^2+^-selective channel and a store-operated Ca^2+^ channel and leads to vesicular fusion. In mud crab, progesterone interacts with scyreprocin on the sperm surface, thus inducing Ca^2+^ influx mediated by SCY2 and initiating sperm AR. The AR molecular basis of mud crab *Scylla paramamosain* revealed in the present study is different from that of other species.

**Table 1 ijms-23-03373-t001:** Antimicrobial Activity of rScyreprocin, rSCY2 and rScyreprocin/rSCY2.

Microorganisms	rScyreprocin		rSCY2		rScyreprocin/rSCY2	
	MIC ^a^ (μM)	MBC ^a^ (μM)	MIC (μM)	MBC (μM)	MIC (μM)	MBC (μM)
*Pseudomonas putida* isolate X1	<0.5	2–4	6.25–12.5	>50	0.5–1	2–4

^a^ MIC and MBC were presented as an interval [A]–[B]: [A] was the highest concentration tested with visible microbial growth, while [B] was determined as the lowest concentration without visible microbial growth (*n* = 3).

## Data Availability

Not applicable.

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
