# Peer review of "Two Male-Specific Antimicrobial Peptides SCY2 and Scyreprocin as Crucial Molecules Participated in the Sperm Acrosome Reaction of Mud Crab Scylla paramamosain"

_ijms, 2022, doi:10.3390/ijms23063373_

Round 1

Reviewer 1 Report

This is a continuation of the authors' previous data about the antimicrobials (SCY2 and scyreprocin) in Mud crab seminal plasma and the authors showed their involvement in the acrosome reaction.

 The entire research is fine however some of the fluorescent images are blurry and also require high resolutions.

Please provide the accession numbers for the genes that were studied for qPCR, I cannot find them on PubMed.

Reviewer 2 Report

Sperm acrosome reaction is a necessary physiological process during fertilization. In this manuscript, Yang et al. investigate the expression of two antimicrobial peptides SCY2 and scyreprocin and their effect on crab sperm function. A few questions or suggestions are below from this reviewer.

  1. Figure 1B, Does "T" from female blot also mean testis? What is it?
  2. please indicate the proteins' name missing in some fluorescence images. for example Fig. 1D, Fig. 3A.
  3. Line 142, is it the first time here the rSCY2 and rScyreprocin apprear in result section? If so, please indicate their full names.
  4. All in vivo and in vitro should be italic.
  5. Line 179, scyreprocin and SCY2 are expressed in multiple organelles. What is their potential function in sperm besides AR regulation? Have authors had any evidence or ever tried them?
  6. Does crab sperm express CatSper channel? What is the calcium influx mediator or pathway?
  7. Figure 4B, What is the concentration of PG used here? F-ASW has a much higher AR ratio than that from male? What is the underlying mechanism? What is the PG concentration in spermathecae?

Round 2

Reviewer 2 Report

The reviewer thank authors for taking these comments. The manuscript has been well revised and there is no more concern from this reviewer.